# In Dormant Red Rice Seeds, the Inhibition of Early Seedling Growth, but Not of Germination, Requires Extracellular ABA

**DOI:** 10.3390/plants11081023

**Published:** 2022-04-09

**Authors:** Alberto Gianinetti

**Affiliations:** Council for Agricultural Research and Economics (CREA), Research Centre for Genomics and Bioinformatics, Via S. Protaso 302, 29017 Fiorenzuola d’Arda, PC, Italy; alberto.gianinetti@crea.gov.it

**Keywords:** seed dormancy, seedling growth, red rice, abscisic acid, fluridone, xanthoxal, xanthoxin

## Abstract

The phytohormone abscisic acid (ABA) inhibits seed germination and seedling growth and is required for the inception of dormancy. Xanthoxal (also known as xanthoxin) is the first specific biosynthetic precursor of ABA. In this study, a modified method to produce xanthoxal is described. I tested the ability of either xanthoxal or ABA to reinstate dormancy in dormant red rice seeds whose dormancy was broken by fluridone (an inhibitor of the synthesis of carotenoids and, subsequently, ABA). Xanthoxal was shown to have a stronger inhibitory effect on germination than ABA when exogenously provided. Although this could indicate an additional effect of xanthoxal above that expected if xanthoxal were simply converted to ABA in the seed, alternative hypotheses cannot be excluded. One alternative is that exogenous xanthoxal may be trapped inside the cells to a greater extent than exogenous ABA, resulting in an intracellular level of ABA higher than that reached with a direct application of ABA. As a further alternative, exogenous xanthoxal may interfere with ABA action in the apoplast. In this study, following germination, early seedling growth was delayed only if ABA was applied. This suggests that inhibition of early seedling growth, but not of germination, requires extracellular ABA.

## 1. Introduction

Physiological dormancy is a state of the seed that, if not released, causes the seed not to germinate or to germinate slowly, even in conditions otherwise suitable for germination [1]. Dormancy is induced during seed maturation and can be relieved by the after-ripening of dry seeds or by the stratification of imbibed seeds [1].

“Red rice” is the common name of weedy rices, a heterogeneous group congeneric to crop rice and characterized by a red caryopsis [2]. These rices show various degrees of seed dormancy; as some accessions are fully dormant, red rice has been proposed as a model plant to elucidate the mechanisms of dormancy [3,4].

The plant hormones abscisic acid (ABA) and gibberellic acids (GAs) act antagonistically in multiple physiological processes, and their balance is critical to normal development and stress responses [5]. Specifically, ABA and GAs are antagonistically involved in seed dormancy and germination, with ABA predominating in dormant seeds and GAs in germinating seeds [1,6]. In particular, ABA is known to function in the process of post-germination growth arrest to protect nascent seedlings from an osmotic environment that turns increasingly stressful and may thus threaten seedling survival [7,8]. In such conditions, seedling growth ceases, and, for a short developmental window, the embryo can re-enter the developmental rest state [8]. This protects the seedling for some time, increasing its chances of encountering favourable conditions [6]. In this respect, the role of ABA in preventing early seedling growth is obvious.

ABA, indeed, has many roles in regulating plant growth, development, and the response to various environmental stresses [6]. Specifically, it shows multiple functions in several stages of seed development [6]. ABA synthesis (in the seed embryo) plays a critical role in initiating dormancy; in fact, null mutations of ABA biosynthetic genes cause the breaking of seed dormancy in many species, including *Arabidopsis thaliana* [9,10], *Solanum lycopersicum* [11], and *Nicotiana plumbaginifolia* [12]. In barley, it has been suggested that the synthesis, catabolism, or removal of ABA, as well as sensitivity to ABA, concur to determine seed dormancy [13].

However, the role of abscisic acid (ABA) in maintaining seed dormancy is far from clear [14,15,16]. In this respect, it is worth pointing out that, even though exogenous ABA enters the embryo, it is not able to restore dormancy (more precisely, to prevent pericarp splitting) in seeds whose ABA synthesis is blocked unless very high non-physiological ABA concentrations are used [14]. Likewise, in many species, sensu stricto germination (testa rupture) of non-dormant seeds is not prevented by ABA [17]. In addition, the degree of dormancy does not always correlate with the dry seed ABA concentration [10,18], and, in wheat, after-ripening relieves seed dormancy without altering the dynamics of the ABA metabolism [19]. Moreover, the proteomic and transcriptomic profiles of dormant arabidopsis seeds differ from those of non-dormant seeds treated with ABA to block their germination and growth, indicating that the physiological state of dormant seeds differs from the state of seeds whose germination is blocked by ABA [20,21].

Fluridone, which blocks the synthesis of carotenoids by inhibiting phytoene desaturase [22], has been used in several studies to prevent ABA biosynthesis: this inhibitor causes the increased or hastened germination of imbibed dormant seeds of red rice [14], arabidopsis [23], barley [24], potato [25], and *Nicotiana plumbaginifolia* [26], among others. Interestingly, fluridone also increased germination in *Orobanche minor* [27] and annual ryegrass [15] without decreasing endogenous ABA levels, suggesting that this inhibitor was acting through a different mechanism. Indeed, an unidentified physiological modulator specific to dormancy and depleted by fluridone has been proposed to explain the inability of ABA to revert the promotion of germination by fluridone [14]. A reduction in the level of some carotenoid-derived molecule other than ABA has been specifically hypothesized to account for the incongruencies between the purportedly central role of ABA in maintaining seed dormancy and the findings mentioned above [28]. In this regard, it has been suggested that the immediate precursors of ABA might have intrinsic biological activity [29].

The enzyme 9-*cis*-epoxycarotenoid dioxygenase (NCED) cleaves C_40_ 9-*cis*-epoxycarotenoids (Figure 1A), specifically either 9-*cis*-violaxanthin or 9′-*cis*-neoxanthin (which is the 9-*cis*-epoxycarotenoid typically found in nature), to produce xanthoxal (whose structure is shown in Figure 1B). This reaction is the first committed step and the rate-limiting step in ABA biosynthesis [6,30], and it is necessary to maintain dormancy [10]. Hence, any carotenoid-derived molecule directly affecting dormancy must be produced by the cleavage of 9-*cis*-epoxycarotenoids to xanthoxal by NCED, or by a reaction occurring after it. In the seed, xanthoxal, the first C_15_ precursor of ABA, is translocated from the plastid to the cytosol, where it is converted to abscisic aldehyde and further to ABA [6]. The IUPAC name of xanthoxal (C_15_H_22_O_3_) is (2Z,4E)-5-[(1S,4S,6R)-4-hydroxy-2,2,6-trimethyl-7-oxabicyclo [4.1.0]heptan-1-yl]-3-methylpenta-2,4-dienal (PubChem CID: 5282222), and its exact mass is 250.15689 Da.

As xanthoxal can enter the seed and then undergo all the conversion steps along the biosynthetic pathway to ABA [30,31], this molecule is the most interesting to test. It is worth noticing that though the common name of this molecule is xanthoxin, “xanthoxal” is recommended as the preferred name [32].

Mutants of *Arabidopsis thaliana* defective in ABA biosynthesis have been used to test possible specific effects of ABA precursors; the results show that although the biosynthetic precursors of ABA can trigger ABA responses in physiological assays, they have limited intrinsic bioactivity, and the physiological activity of ABA precursors derives predominantly from their conversion to ABA in planta [30]. That study used arabidopsis seeds whose dormancy had been fully broken [30], and when dormancy is completely removed (or is suppressed by a mutation of an ABA biosynthetic gene), sensitivity to ABA and to other factors repressing germination is drastically reduced [1,14,26,33,34]. Testing the capability of ABA precursors to maintain dormancy in seeds whose dormancy is relieved by the simultaneous application of fluridone is a more specific assay to establish the effect of these molecules on seed dormancy, since high ABA sensitivity, typical of dormant seeds, is retained [14]. This test is the main purpose of the present research, and because prolonged incubation is required to test the effect on germination [14], I developed a modified procedure to produce enough xanthoxal to carry out the assay.

If exogenous xanthoxal were capable of fully restoring dormancy and ABA were not, it could be inferred that xanthoxal or another ABA precursor produced through the bioconversion of xanthoxal in the seed has intrinsic bioactivity in addition to the physiological activity associated with the synthesis of ABA. Thus, in the present study, the capability of xanthoxal to restore dormancy by reverting the dormancy-breaking effect of fluridone is investigated. In order to accomplish this, a procedure is described that, by improving upon previous methods, allows the production of enough xanthoxal to carry out the test.

## 2. Results and Discussion

To obtain a sufficient amount of xanthoxal for the long germination test, several modifications were made to published methods (e.g., [31,35]) for both the extraction of epoxydic xanthophylls (using environmentally acceptable solvents when possible; see [36]) and xanthoxal purification. Thus, although the principle of the method used here to produce xanthoxal (the oxidative cleavage of epoxydic xanthophylls with zinc permanganate) is well established [31], the usage of more recent routine purification procedures, such as solid-phase extraction (SPE), was introduced. As violaxanthin is more abundant than neoxanthin in green tissues (Appendix A) [31,37], it was used for the routine production of *cis*,*trans*-xanthoxal. Once xanthoxal was produced, its identity was confirmed by chemical characterization. Thereafter, it was used for the germination test.

### 2.1. Xanthoxal Characterization

As assessed by High-Performance Liquid Chromatography (HPLC), a mixture of *cis*,*trans*-xanthoxal and *trans*,*trans*-xanthoxal (where the former is the bioactive form) was obtained by the photoisomerization of the *trans*,*trans*-xanthoxal produced from the oxidative cleavage of violaxanthin (Figure 2). The peak at 3.56 min (assumed to be *trans*,*trans*-xanthoxal) was almost the only isomer present prior to UV irradiation. After irradiation, another peak at 3.85 min was observed. Based on the literature, this latter was identified as the bioactive form *cis*,*trans*-xanthoxal [31]. Correspondingly, the oxidative cleavage of neoxanthin (without photoisomerization) produced this peak only, confirming that it corresponds to the *cis*,*trans* isomer of xanthoxal (not shown). This peak has a maximum at 284.5 nm and a (secondary) *cis* peak at 203 nm (c/h ratio 0.28), whereas *trans*,*trans*-xanthoxal has a maximum at 286.5 nm. The features of the UV−visible spectrum of a *cis* isomer are, indeed, a pronounced *cis* peak at wavelengths sharply shorter than λ_max_ and a very small displacement of λ_max_ itself toward shorter wavelengths [38]. Additional smaller peaks appeared at 2.90 and 3.14 min if irradiation was too intense and prolonged (as occurred for the sample shown in Figure 2). They are supposed to be *trans*,*cis*-xanthoxal and *cis*,*cis*-xanthoxal, respectively. As assessed by HPLC, the xanthoxal isomer mixture did not show a noticeable presence of contaminants (Appendix A).

HPLC coupled with Electrospray Ionization Mass Spectrometry (LC-ESI-MS; Figure 3) confirmed that the obtained chemical had the molecular weight expected for xanthoxal (250.16 Da, to which a proton is added during ESI^+^ to form a quasi-molecular ion denoted [M + H]^+^). The oxygen atoms act as proton attractors, with different outcomes: if the proton hydrogenates the hydroxy group, water is then easily released, and a secondary carbocation is produced (*m*/*z* 233.16) that can undergo a resonance effect (not shown); if the epoxide is attacked, it opens easily because of its strong steric strain, and thereby a hydroxy group and a tertiary carbocation at the two involved carbon atoms form, with two possible isomers. The terminal carbonyl oxygen is less susceptible to proton attack because of the stability of the conjugated double bonds in the side chain, which are disrupted by the formation of a carbocation. Thus, the molecular peak (*m*/*z* 251.17) is expected to correspond to two isomeric tertiary carbocations (more stable than secondary carbocations), although a resonance effect with secondary carbocations is possible (not shown). The putative chemical structure of the main fragments obtained from tandem mass spectrometry (MS/MS, wherein the molecular ion is selected and then split into smaller fragment ions by collision-induced dissociation) is proposed (Appendix A) to show that they are closely compatible with what may be expected from the ESI^+^ fragmentation of the xanthoxal molecule (whose ESI^+^ fragmentation pattern was not available in public databases). Thus, based on the absorbance and mass spectral features, and given it was obtained through the oxidative cleavage of violaxanthin (a process that is known to produce xanthoxal [31]), the isomer generated by photoisomerization was identified as *cis*,*trans*-xanthoxal with high confidence.

### 2.2. Germination Test

Xanthoxal was assayed for its ability to maintain seed dormancy in red rice dormant seeds whose physiological dormancy was simultaneously removed by fluridone (Figure 4A). Both 10 µM (±)-ABA (expected to correspond to 5 µM (+)-ABA [39]) and 5 µM *cis*,*trans*-xanthoxal (applied as part of a mixture of *cis*,*trans*-xanthoxal and *trans*,*trans*-xanthoxal) were able to reduce the percentage of seeds that achieved early seedling growth at similar levels (Figure 4B), confirming that both were biologically active.

ABA prevents seedling growth; however, at physiological concentrations, it cannot re-establish seed dormancy; that is, it cannot persistently block germination [14]. The splitting of the seed-covering layers (i.e., the caryopsis coat, in the case of rice kernels) above the embryo is the conventional marker of germination and, therefore, of dormancy breaking [17]. However, some embryo growth (expansion) is necessary even to accomplish the rupture of the covering layers [17], and ABA can therefore delay pericarp splitting, but it cannot prevent it [14]. Exogenous ABA is more effective at a lower pH level [14] due to the ion-trap mechanism that causes the phytohormone to accumulate in the cellular symplast [14,40]. This might occur because of free diffusion across the cell membrane, but the key role of transporters has increasingly been acknowledged [41,42]. Specifically, some transporters of the ABC (ATP-Binding Cassette) family have been shown to suppress arabidopsis seed germination by importing ABA from the endosperm into the embryo [43]. Although the mechanism of action of the ABA transporters is not yet clear, they favour the passage of ABA through the cell membrane, and, therefore, they should generally facilitate the ion-trap mechanism, since the effect of pH on symplastic-apoplastic ABA equilibration is indisputable [41].

Xanthoxal, however, was able to largely prevent pericarp splitting (i.e., the rupture of the seed testa, or, more precisely, of the caryopsis coat; Figure 4A). Even though *cis*,*trans*-xanthoxal was only partially able to restore dormancy (or, more properly, to restrain germination), it was more effective than ABA (Figure 4A). Since xanthoxal is not expected to be physiologically active in the apoplast, its capability to inhibit germination also confirms that xanthoxal can enter the seed and be converted to ABA [30,31,44]. Given that ABA moves into the cells through specific transporters [42,45] and the xanthoxal molecule is similar to ABA, it may be assumed that xanthoxal can be carried inside the cell by the same transporters that carry ABA. However, xanthoxal conformation is quite different from that of ABA [46]; thus, such transporters must not be very specific.

The greater effectiveness of xanthoxal with respect to ABA was, indeed, found by Taylor and Burden [31] and Kepka et al. [30] for the inhibition of germination in non-dormant seeds of *Lepidium sativum* and *Arabidopsis thaliana abi1-1* mutant, respectively. It is worth noticing that, on the one hand, the stronger effect of xanthoxal observed in the present study cannot be due to the concomitant presence of *trans*,*trans*-xanthoxal in the mixture of *cis*,*trans*-xanthoxal and *trans*,*trans*-xanthoxal, since *trans*,*trans*-xanthoxal (non-photoisomerized xanthoxal from violaxanthin) is not effective (Figure 4A; a very small effect was seen at around 7 d, but this was most probably due to ~5% *cis*,*trans*-xanthoxal that was present, perhaps because light cannot be entirely avoided during xanthoxal preparation). In fact, *trans*,*trans*-xanthoxal is not converted to ABA in planta [47]. On the other hand, it is possible that the unnatural (-)-ABA enantiomer has some biological effect [39]; if true, this would make the result obtained with *cis*,*trans*-xanthoxal even more compelling. Thus, the greater effectiveness of xanthoxal over ABA was the most apparent finding of the germination tests.

A second noticeable finding was that, once germination occurred, seedling growth immediately followed if dormant seeds treated with fluridone were also given xanthoxal, but seedling growth was delayed if ABA was applied instead (Figure 4B,C). Thus, the stronger inhibitory effect on germination of xanthoxal compared to ABA seems to confirm a direct effect of xanthoxal on the maintenance of seed dormancy, but the lack of inhibition of seedling growth by xanthoxal revealed unexpected complexity in the response.

The inhibition of seedling growth by exogenous ABA was accompanied by a secondary effect: during early growth, the embryo collar (see [48] for a description) showed some browning, apparently due to a mild superficial necrosis, and the development of rhizoids was largely curbed (Appendix A). This could be interpreted as a sort of physiological disequilibrium, but it is hard to quantify, even more so because rhizoids are ephemeral structures that atrophy in a few days regardless. Browning was much rarer when dormant seeds induced to germinate by fluridone were treated with xanthoxal rather than ABA: as said, once these seeds attained pericarp splitting, they continued to grow into seedlings without delay; these seeds were able to develop, albeit with wide variability, a thick tuft of rhizoids, as commonly happens when only fluridone, and not ABA, is applied (Appendix A). This seems to suggest that the application of ABA might have brought extracellular (apoplastic) ABA to a higher level than that occurring with a normal, physiological equilibration between symplastic and apoplastic ABA. Exogenous xanthoxal did not provoke this effect, probably because it is not biologically active in the apoplast, and the ABA produced in the symplast from exogenously provided xanthoxal necessarily underwent a more physiological equilibration of its apoplastic level according to the ion-trap mechanism [40]. In accordance, no delay in seedling growth was observed for the few seeds that germinated in the untreated control (also at pH 4.4).

Although *cis*,*trans*-xanthoxal appears to be a better molecular effector of seed dormancy than ABA, it might be argued that xanthoxal is more effective at retrieving seed dormancy (more precisely, inhibiting germination) than ABA because of differences in the trans-membrane compartmentation when either one or the other of these molecules is provided exogenously. In fact, symplastic ABA is in equilibrium with apoplastic ABA, as ABA can move in and out of the cells, but this equilibrium depends on the ion-trap mechanism (since only the non-ionized form can cross the membrane [40]), and this kind of equilibrium should, therefore, be substantially different for xanthoxal, which does not have an ionic form. Specifically, since xanthoxal does not ionize in the apoplast, its uptake into the symplast ought to increase with respect to ABA. In the long run, however, if the transporters that import xanthoxal into the cell, possibly the very same that import ABA, were also responsible for exporting these molecules out of the cell, xanthoxal would not be trapped in the cells. In such case, in fact, xanthoxal would be able to move out of the cell to the same extent as it moved in, apart from the xanthoxal that is metabolized to ABA inside the cell. Xanthoxal, however, is not known to be usually exported outside of seed cells. Therefore, different transporters should be responsible for importing and exporting ABA (this is, indeed, a known fact [6,42]), and only the transporters for the import of ABA into the cell should also be able to transport xanthoxal, whereas those for the export of ABA from the cell should not. If this were true, exogenous xanthoxal would indeed be trapped inside the cells much more effectively than exogenous ABA.

In accordance with the proposed hypothesis of the important role of transporters in determining the cellular concentration, and thus the effect, of these kinds of molecules, an ATP-Binding Cassette (ABC) transporter, MtABCG20, was shown to function as an ABA exporter in germinating seeds of *Medicago truncatula*, and the germination of the loss-of-function mutant, *mtabcg20*, was more sensitive to ABA due to the impairment of ABA export [49]. Of course, this is possible because, even though there are many ABA transporters, each plays a specific role under a given condition and in a specific tissue, so that they are not completely redundant [45].

As said, once xanthoxal is transformed into ABA in the cytoplasm, symplastic-apoplastic ABA equilibration occurs [40,41]. Xanthoxal is indeed rapidly converted to ABA in planta [47], as its conversion is catalysed by enzymes that are constitutively expressed in plant tissues [50] (p. 225). Thus, at equilibrium, ABA derived from (the cellular conversion of) exogenous xanthoxal is expected to block seedling growth at the same level as exogenous ABA, given that the same amount of biologically active (S)-ABA would ultimately be provided, in the two treatments, to the symplast as well to the apoplast. A similar level of inhibition of seedling growth was indeed observed for the two conditions; however, in xanthoxal-treated seeds, seedling growth was prevented only if pericarp splitting (germination) was prevented too (Figure 4A,B). In ABA-treated seeds, on the other hand, seedling growth was delayed following germination (Figure 4C). Some mechanism, therefore, hampers a comparable symplastic-apoplastic ABA distribution across the two treatments.

Thus, the stronger inhibition of germination (assessed using pericarp splitting as a visual marker) by exogenous xanthoxal seems to suggest two possibilities. Either the application of xanthoxal causes a higher ABA concentration inside the cells than if ABA itself is applied (that is, the equilibrium between symplastic and apoplastic ABA is not reached because, for example, ABA is continuously produced inside the cell faster than it is exported to the apoplast) or, instead, xanthoxal has an effect per se (as suggested long ago [51]), that is, it is recognized by some intracellular receptor as a biologically active molecule. On the basis of the ion-trap mechanism, however, the former hypothesis seems doubtful, as the equilibration between intracellular and extracellular ABA ought not to be a slower process than the ABA biosynthesis from xanthoxal. Nevertheless, this might occur if the equilibrium between symplastic and apoplastic ABA is not reached because some limiting mechanism operates to restrain ABA import but not xanthoxal import when intracellular ABA concentration reaches some threshold. On the one hand, this additional hypothesis is more consistent with the above-mentioned observation that the application of ABA might bring apoplastic ABA to a higher level than that occurring with a physiological equilibration between symplastic and apoplastic ABA. On the other hand, the existence of such a mechanism has never been demonstrated. In addition, this hypothesis also requires that the ABA produced from xanthoxal in the symplast is, in turn, prevented from reaching the same apoplastic level as when ABA is provided. Although this could be explained by a quick intracellular catabolism of any ABA excess, the present findings reveal that something is missing in our present understanding of the regulation of ABA response/transport/catabolism. As for the latter hypothesis (namely, that xanthoxal has an effect per se), since the biological effects of xanthoxal are very close to those of ABA, and the xanthoxal molecule is similar to ABA, one might suppose that xanthoxal acts on ABA receptors, as noticed above for ABA transporters. However, conformational differences between xanthoxal and ABA [46] make this assumption even more speculative for receptors than for transporters. In fact, xanthoxal is expected to have a poor affinity for PYL receptors, given that the carboxylic moiety has a role in forming a stable complex with the receptor to trigger a signal [46]. Dedicated intracellular xanthoxal receptors, on the other hand, are expected to cause a complete reversal of the effect of fluridone, with full restoration of dormancy in seeds treated with xanthoxal, given that they were provided with a relatively high xanthoxal amount. Such a complete reversal, however, did not occur. Thus, both hypotheses seem to fall short of providing a satisfactory explanation for the observed results, at least based on the present knowledge. Yet, our knowledge of PYR/PYL ABA receptors is still limited, and relevant differences in their properties have been observed among different species [52].

In any case, the diverging effects of applied xanthoxal on germination and seedling growth indicate that different receptors are involved in the control of germination (pericarp splitting) and early seedling growth (rootlet or coleoptile ≥ 1 mm). An easy explanation for the observed separation of the two effects (namely, strong inhibition of germination but no inhibition of seedling growth following germination) is that hormone receptors negatively controlling germination and seedling growth are located in the symplast and the apoplast, respectively. ABA, indeed, can be perceived at both intracellular and extracellular sites [53]. A divergence in the response to exogenous ABA between radicle emergence and seedling growth has also been observed for the suppression of ABA inhibition by low concentrations of sugars in arabidopsis [54]. In that case too, either an alteration of ABA availability at one site of ABA perception or of a specific signalling pathway was suggested to be responsible for the two independent responses [54]. In the present case, as xanthoxal is physiologically converted to ABA inside the cells, a different localization of ABA receptors affecting radicle emergence and seedling growth appears to be the most obvious explanation for the divergent responses observed at these two stages when either xanthoxal or ABA were exogenously provided.

As a further alternative hypothesis to explain why exogenous xanthoxal inhibited pericarp splitting but not seedling growth, which immediately followed whenever pericarp split occurred, whereas exogenous ABA delayed seedling growth after pericarp splitting, it might be supposed that ABA effectiveness was lower at the site of growth control (possibly the apoplast) in xanthoxal-treated seeds than in ABA-treated seeds. Assuming that xanthoxal is quickly converted to ABA in the cytoplasm and that a normal equilibrium between symplastic and apoplastic ABA would then be promptly achieved, it could be hypothesized that exogenous xanthoxal would negatively interfere with ABA signalling in the apoplast (where, physiologically, there should not be xanthoxal). Maybe they compete for the same receptors, but xanthoxal does not elicit a response. Even in this case, the hypothesis that hormone receptors repressing germination and seedling growth are located in the symplast and apoplast, respectively, would provide a simple explanation for the diverging effects of exogenous xanthoxal vs. exogenous ABA on germination and seedling growth. So, at present, there are several possible explanations for the greater efficacy of xanthoxal over ABA, but whatever the correct explanation is, an apoplastic localization of ABA receptors for the inhibition of seedling growth is highly consistent with the present findings.

Further research involving a more detailed analysis of xanthoxal and ABA levels in the apoplast and symplast is necessary to disentangle this matter and answer all the questions raised by the present study.

## 3. Materials and Methods

### 3.1. Chemicals

Stock solutions were prepared by dissolving fluridone (Duchefa, Haarlem, The Netherlands) and racemic (±)-ABA (Sigma, St Louis, MO, USA) in dimethylsulphoxide (DMSO; Sigma, St Louis, MO, USA) to a concentration of 0.1 M. Xanthoxal (a mixture of geometric isomers produced as detailed below) was dissolved in DMSO to a concentration of 0.1 M (total xanthoxal). If the DMSO volume required to prepare a new xanthoxal stock was <200 µL, the stock volume was brought to 200 µL with a 20% (weight for volume of water) aqueous solution of polyethylene glycol (PEG) 6000 (BDH, Poole, UK). Pre-dilution to such minimum volume was necessary to wash all the xanthoxal off the walls of the vial. These stock solutions were stored at +5 °C, wrapped with aluminium foil; pre-diluted xanthoxal stock was prepared just before the experimental run (with a freshly dissolved PEG solution). The seed incubation solution was buffered to pH 4.4 with 20 mM 1,2,3,4-butanetetracarboxylic acid (BTCA; Aldrich, Steinheim, Germany) [14].

### 3.2. Xanthoxal Production

#### 3.2.1. Extraction of Xanthophylls

Freshly collected maize leaves (about 300 g) were immediately immersed in 50:50 ethanol/ethyl acetate (600 mL) while protecting them from direct sunlight. Ten grams of sodium bicarbonate was gradually added together with the leaves into an amber bottle. A needle through the cap prevented pressure build-up. After overnight extraction, the liquid was percolated, and the leaves were extracted again with ethyl acetate overnight. This was repeated for a third extraction. The extracts were progressively combined while the volume was reduced by means of a vacuum rotary evaporator (Heidolph Instrument, Kelheim, Germany) at ≤30 °C. When evaporation slowed down, 50 g K_2_HPO_4_ was added. Whereas ethanol, which is a mildly chaotropic agent, was used in the first extraction to obtain a single phase from the organic (ethyl acetate) and the aqueous phases, K_2_HPO_4_, which is a strongly kosmotropic salt, was subsequently added to again separate (salting out) the two phases [55]. The volume was further reduced, and the liquid was discharged when it appeared to be dark yellow (in leaf extracts, the aqueous phase is yellow because of flavones [56]) as the chlorophylls sedimented on the rotating flask wall (indicating that the liquid was mostly water, since chlorophylls remain fully soluble when there is ≥15% ethanol).

#### 3.2.2. Liquid-Liquid Partition of Xanthophylls

The sediment was re-suspended in 200 mL 90:10 hexane/ethyl acetate and transferred to a separatory funnel. Then, 200 mL 80:20 methanol/water was added, the mixture was thoroughly mixed, and two (green) layers were left to separate. Epoxydic xanthophylls partitioned to the lower methanolic phase, and the upper one was therefore discharged. The methanolic phase was washed two more times with 200 mL 90:10 hexane/ethyl acetate to remove most chlorophylls. A small excess K_2_HPO_4_ (about 1 g) was then added to the methanolic phase. A small volume of aqueous phase (which is yellow because of flavones [56]) and the undissolved salt were discharged. A small amount of sodium sulphate was added, and the liquid was slowly percolated through filter paper. The organic phase was transferred to a flask, and 20 mL ethanol and a small excess of K_2_HPO_4_ were added. The volume was reduced with a rotary evaporator, but complete drying was avoided. Then, 100 mL 80:20 hexane/ethyl acetate was added, and part of the sediment was re-dissolved. The liquid was transferred to a new flask, a small amount of sodium sulphate was added, and the organic phase was slowly percolated through filter paper. The extract was stored under N_2_. The analytical determination of carotenoids was performed with normal-phase HPLC (see Section 3.3.1. HPLC Analyses).

#### 3.2.3. Purification of Violaxanthin and Neoxanthin with Solid-Phase Extraction

To further remove salts, chlorophylls, and most lutein, solid-phase extraction (SPE) was carried out with a silica column (60 mL, 10 g silica-Discovery^®^, Supelco; under suction). The column was conditioned (and deacidified) with 20 mL 80:20 hexane/ethyl acetate containing 1% triethylamine (TEA). After loading the sample (about 100 mL, in 80:20 hexane/ethyl acetate), violaxanthin was eluted with 100 mL di 50:50 hexane/ethyl acetate containing 0.05% TEA. Some chlorophylls eluted first and were discharged. Neoxanthin was eluted with 70 mL ethyl acetate. Normal-phase HPLC (see Section 3.3.1. HPLC Analyses) was used to check the purity of the two epoxydic xanthophylls. If they were satisfactorily pure, they were dried with a rotary evaporator, re-suspended in 1–2 mL acetone, collected in a small vial, and dried under N_2_ flow. The weight of each xanthophyll was calculated by difference, and the two vials were stored at +5 °C.

To remove lutein from the violaxanthin fraction, SPE with a 2 g C18 column (Discovery^®^, Supelco; under suction) was performed. The column was pre-treated with 4 mL acetone, deacidified with 8 mL 25 mM K_2_HPO_4_, washed (de-salted) with 10 mL water, and conditioned with 7 mL 5:2 acetone/water containing 0.2% TEA. The dried violaxanthin fraction was dissolved in 7 mL 5:2 acetone/water containing 0.2% TEA, and it was then loaded. The column was then washed with 14 mL 5:2 acetone/water containing 0.05% TEA. Violaxanthin was eluted with 7 mL 6:1 acetone/water. The fraction was dried with a rotary evaporator, re-suspended in 1–2 mL 80:20 hexane/ethyl acetate (acetone could dissolve some residual salt), and transferred to a glass vial. Purity was checked in normal-phase HPLC (see Section 3.3.1. HPLC Analyses). If satisfactory, the sample was dried and stored as described above.

If some chlorophyll contamination persisted in a xanthophyll fraction, further SPE was carried out with a 1 g silica column (Discovery^®^, Supelco; under suction) as follows. The dried xanthophyll sample was dissolved in 2 mL acetonitrile (ACN) containing 0.25% TEA. The column was pre-treated with 2 mL ACN, deacidified with 5 mL 20% ACN in water containing 0.2% TEA and 2% K_2_HPO_4_ (*w*/*v*), and then conditioned with 10 mL ACN. After loading the sample, the epoxydic xanthophyll was eluted with ACN. Purity was then checked in reversed-phase HPLC with ACN/water (see Section 3.3.1. HPLC Analyses). If satisfactory, the epoxydic xanthophyll fraction was partitioned three times to diethyl ether by adding a small volume of diethyl ether, mixing, and then adding three times as much water as the volume of the ACN (i.e., the original volume of the eluted fraction). It was finally dried and stored in a small vial as described above. For long storage, epoxydic xanthophylls were kept at −20 °C.

#### 3.2.4. Production of Xanthoxal by Oxidation of Epoxydic Xanthophylls

Xanthoxal was produced by the oxidative cleavage of epoxydic xanthophylls with zinc permanganate [31]. To minimize the oxidative degradation of xanthoxal once formed, equimolar concentrations of xanthophyll and permanganate were used (assuming that a xanthophyll molecule should be cut once, and a molecule of permanganate is required to cleave one double bond). As part of the epoxydic xanthophyll remained un-cleaved after the oxidation, additional cycles of xanthophyll separation/oxidative cleavage were carried out. In the case of violaxanthin, which is symmetric and can therefore produce two xanthoxal molecules, C_25_-epoxy-apocarotenal (the molecule that is produced together with xanthoxal with a single oxidative cut) and other epoxy-apocarotenals were combined with un-cleaved violaxanthin and subjected to oxidative cleavage again. Two to four oxidative cleavages were usually necessary to cleave all the violaxanthin and the allenic-apocarotenals (chiefly, C_25_-epoxy-apocarotenal) derived from it. Two oxidative cleavages were often enough for neoxanthin, as the C_25_-allenic-apocarotenal and other allenic-apocarotenals cannot produce xanthoxal.

For the first oxidative cleavage (100 µmoles of each compound), 60 mg epoxydic xanthophyll (from multiple collections) was dissolved in 14 mL acetone in a 50 mL Falcon tube, 18.3 mg zinc acetate was dissolved in 1 mL water, and 15.8 mg KMnO_4_ was dissolved in 1 mL water. To the acetonic xanthophyll solution, 4.7 mL 0.2 M BTCA buffer pH 6 buffer, the permanganate solution (1 mL), and 300 µL of the zinc acetate solution were added. If smaller amounts of epoxydic xanthophyll were available (particularly in the case of neoxanthin), the reagent volumes were scaled down proportionally (in a 15 mL Falcon tube, if epoxydic xanthophyll was <20 mg). The mix was intermittently mixed by vortexing at slow speed for 10 min. One volume of diethyl ether was added to extract xanthoxal, apocarotenals, and residual xanthophyll. After thoroughly mixing, the mix was centrifuged for 5 min at 1000× *g*. The yellowish organic (upper) phase was transferred to a 250 mL round-bottom flask. The aqueous (lower) phase was re-extracted with diethyl ether by mixing and centrifuging as above. This second organic extract was united with the first one, and the sample was slowly dried with a rotary evaporator.

To separate xanthoxal, the dry sample was re-suspended in 4 mL ACN, 6 mL water was added, and SPE was performed with a 2 g C18 column (Discovery^®^, Supelco; under suction). The column was pre-treated with 4 mL ACN, conditioned and deacidified with 10 mL 40% ACN containing 0.5% K_2_HPO_4_. The xanthoxal fraction was collected from sample loading (in 40% ACN) up to elution with 10 mL 40% ACN. Residual xanthophyll and C_25_-epoxy-apocarotenal were eluted with 6 mL ethyl acetate and dried with a rotary evaporator.

Second, third, and even fourth oxidative cleavages were performed on the residual xanthophylls with the same procedure and doses of permanganate and zinc acetate as above (assuming that all the cleaved violaxanthin produced C_25_-epoxy-apocarotenal). When all the useful cleavages had been practically achieved (as judged from the colour intensity of the residual xanthophyll fraction or by normal-phase HPLC; see Section 3.3.1. HPLC Analyses), the xanthoxal fractions obtained from the various oxidation steps were combined. The overall yield of xanthoxal was quite low, roughly 10%, since there are several double bonds at which oxidative cleavage can occur, but only two in violaxanthin and one in neoxanthin are conducive to xanthoxal production, provided that double bonds internal to the xanthoxal residue are not cleaved as well.

#### 3.2.5. Purification of Xanthoxal with Reversed-Phase SPE

To concentrate xanthoxal and remove most of the unwanted xanthophyll fragments as well as colloidal manganese oxides, three volumes of water was added to the xanthoxal solution (in 40% ACN), and SPE with a 2 g C18 column (Discovery^®^, Supelco) was carried out (under suction). The column was pre-treated with 4 mL ACN, conditioned and deacidified with 10 mL 10% ACN containing 0.5% K_2_HPO_4_. After loading the sample (in 10% ACN), the column was washed with 10 mL 10% ACN containing 0.5% K_2_HPO_4_ (to remove any residual manganese and zinc cations) and then with 20 mL 25% ACN. Xanthoxal was eluted with 10 mL 40% ACN. Different fractions were collected (typically three, of 2, 6, and 2 mL, respectively) and analysed with reversed-phase HPLC with 40% ACN isocratic flow (see Section 3.3.1. HPLC Analyses). Fractions containing xanthoxal were combined. The *trans*,*trans*-xanthoxal could still display a yellowish colour due to traces of polar contaminants, presumably long carotene dialdehydes derived from violaxanthin.

The xanthoxal sample (in 40% ACN) was diluted with 1/3 of water (*v*/*v*) to 30% ACN, and then partitioned to diethyl ether three times. The latter was then dried almost completely (in a glass vial) under N_2_ flow.

#### 3.2.6. Photoisomerization

Oxidative cleavage of neoxanthin (which is naturally present as 9′-*cis*-neoxanthin) directly produces the bioactive *cis*,*trans*-xanthoxal form; thus, that neoxanthin is the preferred natural molecule for producing bioactive xanthoxal. Nevertheless, violaxanthin is much more abundant than neoxanthin in green tissues; it is therefore a better source for producing *cis*,*trans*-xanthoxal by oxidative cleavage [31]. However, all-*trans*-violaxanthin is the isomer typically found in leaves; therefore, *trans*,*trans*-xanthoxal is obtained by its oxidative cleavage. The bioactive geometric isomer can then be obtained by photoisomerization [31], since photochemical activation produces a transient excited state that loses the double-bond character and therefore allows rotation around the bond [57]. The double bond can then re-form upon return to the ground state. As the reaction is photoreversible, a mixture of the geometric isomers is produced that tends to an equilibrium or photostationary state, wherein the *cis*,*trans* isomer (which has a Dreiding energy about 1 kJ/mol higher than the *trans*,*trans* isomer) reaches its maximum equilibrium proportion [57]. Preliminary experiments showed that a maximum proportion of the peak area of *cis*,*trans*-xanthoxal of slightly more than half (about 3/5) that of *trans*,*trans*-xanthoxal could be obtained, in agreement with previous observations [32]. As a *cis* isomer has a slightly smaller peak absorbance than a *trans* isomer [38], it is estimated that a maximum proportion of *cis*,*trans*-xanthoxal of about 40% (of the total xanthoxal) can be obtained by photoisomerization of *trans*,*trans*-xanthoxal, less than originally inferred by Taylor and Burden [31].

For photoisomerization, *trans*,*trans*-xanthoxal was dissolved with 3 mL ACN in a 20-mL glass vial (to reduce UV absorption) and put under blacklight (light source: Philips TL Mini Blacklight Blue TL 6W BLB 1FM, emission at 350–400 nm) for 1–1.5 h (at a distance of about 2 cm from the lamp). Some isomerization at the second double bond could occur during the process, especially if irradiation was too intense (Figure 2B). The reaction time was chosen as the time required for the area of the *cis*,*trans*-xanthoxal peak to become at least half that of the *trans*,*trans*-xanthoxal in reversed-phase HPLC, but before noticeable isomerization at the second double bond occurred (<4% of the total xanthoxal). Thereafter, the sample was diluted with water to 30% ACN, and partitioned to diethyl ether three times. It was finally transferred into a 50-mL round-bottom flask and dried with the rotary evaporator.

#### 3.2.7. Final Purification of Xanthoxal with Normal-Phase SPE

SPE with a 1 g diolic column (Discovery^®^, Supelco) was carried out (under suction) for the isomer mixture obtained from violaxanthin after photoisomerization as well as for *cis*,*trans*-xanthoxal produced from neoxanthin (without photoisomerization). The xanthoxal sample was dissolved in 2 mL 40:60 ethyl acetate/hexane. The column was deacidified with 5 mL hexane containing 0.5% TEA and conditioned with 10 mL 40:60 ethyl acetate/hexane. After loading the sample, the column was washed with 1 mL 40:60 ethyl acetate/hexane. Xanthoxal was then eluted with 12 mL 40:60 ethyl acetate/hexane: six fractions, 2 mL each, were collected in glass vials. Any residues of butylated hydroxytoluene (coming from diethyl ether, wherein it is commonly present as a stabilizer) and carotenoid fragments were removed with this step. Purity was tested in normal-phase HPLC with an isocratic 70:30 ethyl acetate/hexane flow (as described in the next section). The xanthoxal fractions were united and dried under N_2_ flow; the amount of xanthoxal was calculated by the difference compared to the weight of the empty vial, and the sample was stored at +5 °C. For long storage, xanthoxal was kept at −20 °C. On average, about 0.5–2 mg of total xanthoxal was obtained per kg of fresh maize leaves; this means about 0.2–0.8 mg *cis*,*trans*-xanthoxal was produced (after photoisomerization).

### 3.3. Chemical Analyses

#### 3.3.1. HPLC Analyses

For the determination of xanthophylls and other carotenoids, normal-phase HPLC was used [58] with a Beckman-System Gold chromatograph (Beckman Instrument Inc., Brea, CA, USA). Twenty µL of each sample (diluted 1:10 with 80:20 hexane/ethyl acetate) was injected and eluted through a 250 mm × 4 mm 5 µm Lichrospher Si60 column (Merck KgaA, Darmstadt, Germany) with a guard column of the same stationary phase. The mobile phase used was a gradient of solvent A (hexane containing 0.05% TEA) and solvent B (ethyl acetate containing 0.05% TEA) with a flow rate of 1 mL/min. The following steps were used: 20% B for 1 min, from 20% to 70% B over 2 min, to 76% B over 6 min, to 100% B over 1 min, constant 100% B for 4 min, to 20% B over 1 min, 20% B for 3 min (18 min total). Absorbance was measured at 436 nm [59] with a photodiode array detector. The different carotenoids (Appendix A) were identified based on their spectroscopic characteristics [37,60].

When epoxydic xanthophylls were dissolved in ACN, reversed-phase HPLC was executed with an Agilent 1200 chromatograph (Agilent Technologies, Waldbronn, Germany). Ten µL was loaded with an auto injector and eluted through a 150 × 4.5 mm 5 μm Zorbax Eclipse XDB-C18 analytical column (Agilent Technologies, Waldbronn, Germany) with a guard column. Isocratic elution was performed with 99.8% ACN containing 0.05% TEA and 0.2% aqueous solution of 3% formic acid for 8 min (1mL/min, 25 °C). Absorbance was measured at 450 nm and 285 nm.

To properly distinguish xanthoxal isomers, reversed-phase HPLC was carried out with the Agilent 1200 system, using isocratic elution (1 mL/min, 25 °C) with 40% ACN containing 0.02% TEA. Absorbance was measured at 285 nm for 5 min.

To assess xanthoxal purity after the final SPE, normal-phase HPLC was used with the Beckman-System Gold system. Twenty µL of each sample (diluted twenty times with 70:30 ethyl acetate/hexane) was injected. Isocratic elution (1 mL/min) with 70:30 ethyl acetate/hexane (containing 0.05% TEA) was followed at 280 nm for 7 min. The retention time for the largely overlapping *trans*,*trans*-xanthoxal and *cis*,*trans*-xanthoxal isomers was 4.1 min.

#### 3.3.2. Mass Spectrometry

Mass Spectrometry (MS) analysis was carried out on a mixture of the two geometric isomers obtained from violaxanthin (after photoisomerization) with a Triple-Stage Quadrupole mass spectrometer with an electrospray ionization (ESI^+^) source (TSQ Quantum, Thermo Scientific, San Jose, CA, USA). A Surveyor HPLC (shortly, LC) device with a 125 mm × 4 mm × 5 µm C18 column was connected to the spectrometer for LC-(PDA)-ESI-MS and tandem mass spectrometry (MS/MS). 0.1% formic acid in water and ACN were used as solvent A and solvent B, respectively. Elution consisted of a gradient from 5% B to 95% B over 25 min. Absorbance was followed (for 28 min) in the 200–400 nm range with a channel at 285 nm. For LC-ESI-MS, “full mass” signals (every 100 msec) in the 200–400 *m*/*z* range were initially analysed to ascertain the absence of other noticeable masses, and the range was then restricted to 220–260 *m*/*z*. The only two strong signals observed, namely, the molecular peak at 251 *m*/*z* [M+H]^+^ and its dehydrated form with 233 *m*/*z* (Figure 3), were then fragmented in MS/MS. In LC-ESI-MS/MS, the most abundant peak at each microscansion is fragmented with a fixed collision energy (default 30 eV); for xanthoxal, this resulted in a complex fragmentation pattern (Appendix A). Hence, direct infusion ESI-MS/MS was performed (with xanthoxal dissolved in methanol), and the collision energy was modulated to a lower level such that the molecular peak did not completely disappear (Appendix A).

### 3.4. Chemical Editor

MarvinSketch and MarvinView (version 19.21.0, 2019; ChemAxon, http://www.chemaxon.com; last accessed on 19 November 2019) were used for drawing, displaying, and characterizing the chemical structures and their 3D conformational isomerism, including the conformer Dreiding energy. The latter represents the relative potential energy of the 3D structure (conformation) of a molecule calculated by using the Dreiding force field after automatic optimization of the molecular structure. The Dreiding energy of the lowest-energy (more stable) conformer was used as an aid to choose the putative structure of the main positively charged fragments, based on their stability, among several potential candidate structures inferred from the mass spectra.

### 3.5. Seed Materials and Experimental Setup

A straw-hulled red rice genotype originally found in a paddy close to Vercelli (located in a rice-growing area of the Po Valley, Italy), and previously used for other studies [14,16,48,61], was grown in a greenhouse at Fiorenzuola d’Arda (Italy). The seed grains (botanically, the rice dispersal units are spikelets) were harvested when showing shattering capability and dried for 1 d at 35 °C [48]. Dormant red rice spikelets (germination ≤ 3% in water) were stored at −15 °C till use [16]. Naked (dehulled) caryopses were prepared by manually dehulling the spikelets prior to the start of the experimental run [16,48]. Although the naked rice grain is botanically a caryopsis, for the sake of simplicity it is referred to as a “seed”, in a broad sense, throughout this paper.

Dehulled dormant red rice caryopses produce superoxide for about two days following imbibition [62], and such reactive oxygen species can easily degrade xanthoxal, since it reacts with conjugated dienes [63] (p. 72) so that carbonyl carotenoids, even short-chain ones, easily take up electron pairs [64]. For that reason, and also because of the quick reactivity of the aldehyde group with peroxyl radicals [65], the seeds were pre-imbibed at 30 °C for 2–10 days in plastic Petri dishes (90 mm diameter; VWR International, Milano, Ialy) on two filter paper discs (90 mm diameter; Whatman grade 1, GE Healthcare Life Sciences, Little Chalfont, UK), with 5 mL of water, before testing. Any seed that germinated during pre-imbibition was excluded from the subsequent trials. All the seeds used in a given experimental run were pre-imbibed for the same period of time. The effect of chemicals was then tested by transferring the seeds to 100 mL Erlenmeyer flasks (capped with aluminium foil), enclosed in a humidity box at 30 °C. The seeds were placed on a Whatman 3 MM (GE Healthcare Life Sciences, Little Chalfont, UK) paper disc (4 cm diameter, cut out by hand) with 2 mL of the incubation solution [14]. For each treatment, two to six (typically three) replicate flasks, with 12 caryopses each, were used. All incubation solutions were buffered at pH 4.8 (the pK_a_ of ABA is 4.75 [40]) with 20 mM BTCA [14]. Acidic conditions enhance ABA effectiveness [14,40] and reduce the production of superoxide by seeds [62]. In the control (dormant seeds without bioactive chemicals), the seeds were incubated in the buffer solution. Treatments with bioactive chemicals also included: 10 µM fluridone, 10 µM fluridone + 10 µM (±)-ABA [14], or 10 µM fluridone + 5 µM *cis*,*trans*-xanthoxal (applied as a mixture of isomers from violaxanthin). The concentrations of fluridone and ABA were the same as used by Gianinetti and Vernieri [14]. Immediately prior to adding fluridone, the diluted buffer solution and the fluridone stock were pre-warmed at about 37 °C to improve the dissolution of fluridone in the aqueous solution. The concentration of *cis*,*trans*-xanthoxal was fixed at 5 µM for optimal comparison: because racemic (±)-ABA is a mixture of S(+) and R(−) forms in approximately equal amounts [39], whereas the naturally occurring form is (S)-*cis*-ABA, also known as (+)-*cis*,*trans*-abscisic acid, the biologically effective xanthoxal isomer, (S)-*cis*,*trans*-xanthoxal (the S structure is the natural structure of ABA precursors), was applied at the same expected concentration as the biologically effective ABA isomer, that is, half the concentration of (±)-ABA. The mixture of isomers produced from violaxanthin was used in the xanthoxal test, and the concentration of *cis*,*trans*-xanthoxal was kept at 5 µM. As very small amounts of xanthoxal were commonly handled, the concentration of *cis*,*trans*-xanthoxal in the incubation solution was adjusted to the desired concentration (5 µM) on the basis of the corresponding peak area obtained from reversed-phase HPLC. Based on a large sample preparation, it was estimated that, with the described method and using an injected volume of 3 µL, one peak unit area corresponded to a *trans*,*trans*-xanthoxal concentration of about 62 µg/L. Although for *cis*,*trans*-xanthoxal this value ought to be slightly higher, the proportion of the *cis*,*trans*-xanthoxal isomer in the isomer mixture produced from violaxanthin was directly calculated from the area of the HPLC peak, without considering differences in absorbance among isomers. The caryopses were transferred to new flasks with fresh incubation solutions three times a week (on Monday, Wednesday, and Friday). The incubation solutions containing the bioactive chemicals were prepared the day prior to first using them, in amounts enough for one week; they were then stored at +5 °C. Preliminary HPLC checks showed no degradation of the chemicals in the stored aqueous solutions over this time (the solution was quite stable even for several weeks).

Two stages of germination were recorded (at every change of flask): (i) pericarp splitting, the first visible sign of germination in red rice [14], which in cereals is more precisely defined as the rupture of the caryopsis coat [48] and is more generally known as the rupture of the seed testa [17]; and (ii) the first growth stage (S1 [66]), recorded when rootlets or coleoptiles were ≥1 mm (minimal visible seedling growth). Stage S1 includes some early seedling growth that makes germination easier to see [67]. Pericarp splitting was determined, with the aid of a magnifier, as the opening of the red caryopsis coat covering the swelling embryo axis into two lips, disclosing the underlying tissues. Seedlings attaining growth stage S1 were always discharged after recording. At the end of each test, non-germinated caryopses were transferred to Petri dishes with two discs of filter paper and 4 mL of water to assess their viability. In fact, the dormancy-breaking effect of fluridone persists well after removal of the exogenous supply, whereas the germination-delaying effect of 10 µM ABA does not, so that viable seeds quickly germinate when the incubation solution is replaced with water [14]. Xanthoxal, which is metabolized to ABA, also loses its effectiveness when no longer provided exogenously. Caryopses that did not attain growth stage S1 in one week of incubation at 30 °C after being transferred to water were considered not viable. The difference between observed percentages of growth in stage S1 and germination was used as an indicator of the delay of seedling growth after germination was attained, i.e., as an indirect measure of the apparent delay of germination. The rationale for this indicator is that if seedling growth is quick, at the time of recording, only a few seeds are observed at the pericarp splitting stage before seedling growth is apparent (as the time between the two stages is very short). As the time between the two stages increases, a higher proportion of seeds is expected to be detected at the pericarp splitting stage before they display seedling growth.

### 3.6. Statistical Analysis

Statistical analysis of the germination data was performed with the GLIMMIX procedure of the SAS^®^ 9.4 software (SAS^®^ OnDemand for Academics; SAS Institute Inc., Cary, NC, USA) according to a conditional generalized linear mixed model using the probit link function and Laplace approximation of maximum likelihood [68]. The treatment effect was modelled through time as a spline; this was because of the multiple observation times, which differed across experimental runs (the experiment was, in fact, split into many incomplete experimental runs, each typically including two treatments, and each treatment was repeated over three to six experimental runs), and also depended on the day of the week the individual experimental run started. The interaction between treatment and experimental run was modelled as a random factor, and the Petri dish effect was modelled as a random factor with first-order autoregressive covariance structure [68]. Additional information about the statistical analysis, including the SAS code, is given in Appendix B.

## 4. Conclusions

In red rice seeds whose dormancy is broken by fluridone, exogenously applied xanthoxal is more effective at inhibiting germination than exogenous ABA, but it is ineffective at further delaying early seedling growth once germination (pericarp splitting) is achieved. It remains to be established which of three suggested hypotheses explains this result. The first possibility is that the former effect is a consequence of xanthoxal being a non-ionic compound whilst ABA is a dissociable acid, such that exogenous xanthoxal enters the cells easier than ABA; however, this hypothesis requires some mechanism that prevents the achievement of a corresponding equilibrium between symplastic and apoplastic ABA when either xanthoxal or ABA is exogenously provided. The second possibility is that xanthoxal carries out a competitive inhibition of ABA action in the apoplast, thus suppressing the capability of ABA to delay seedling growth. The final possibility is that xanthoxal plays a specific role in dormancy, that is, it specifically blocks germination rather than seedling growth; however, like for the first possibility, a further mechanism must be invoked to explain the inhibition of seedling growth by exogenous ABA only. Thus, both the first and the third hypotheses imply the existence of a physiological restraint on ABA export/import, at least when ABA reaches some biologically meaningful threshold. In any case, the contrasting effects of exogenous xanthoxal on pericarp splitting and early seedling growth, compared to exogenous ABA, indicate that different receptors are involved in these two processes. As a tentative explanation, it is here proposed that symplastic receptors are involved in the regulation of seed germination, whereas apoplastic ABA receptors control early seedling growth. It is certain that extracellular ABA was required to inhibit early seedling growth.

## Figures and Tables

**Figure 1 plants-11-01023-f001:**
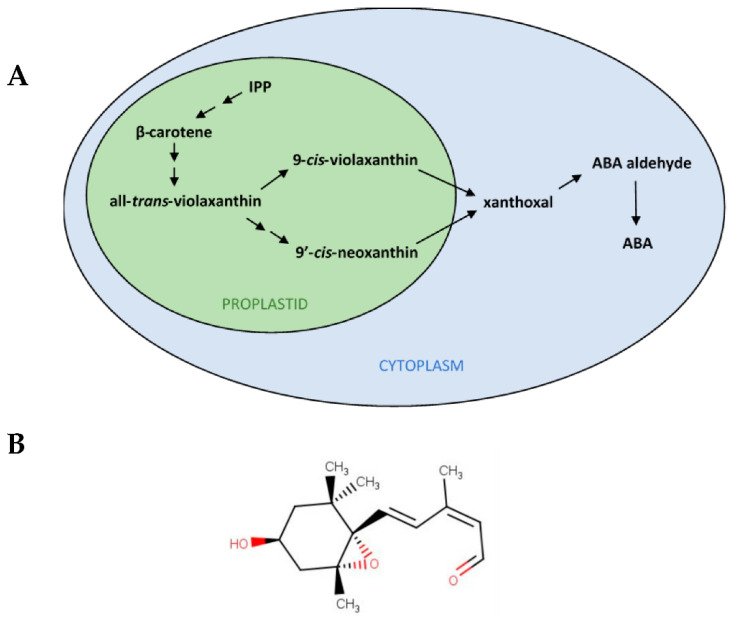
Xanthoxal features. (**A**) Scheme of the abscisic acid (ABA) biosynthesis pathway in plants. ABA synthesis proceeds from the carotenoid pathway, which starts from the precursor isopentenyl pyrophosphate (IPP), a C_5_ (five carbon atoms molecule), and, through β-carotene, a C_40_, produces epoxydic xanthophylls, such as all-*trans*-violaxanthin, still a C_40_. By isomerization, 9′-*cis*-neoxanthin and 9-*cis*-violaxanthin are further obtained, which can be oxidatively cleaved by 9-*cis*-epoxycarotenoid dioxygenases (NCEDs) in the first specifically committed step of ABA synthesis, resulting in the production of xanthoxal, a C_15_. Up to the C_40_ to C_15_ conversion, which is the rate-limiting step of ABA biosynthesis, the pathway takes place in the plastids (proplastids, in seeds). Thereafter, in the cytoplasm, ABA (still a C_15_) is synthesized from xanthoxal through ABA aldehyde. Double arrows indicate multiple biosynthetic steps. (**B**) Chemical structure of *cis*,*trans*-xanthoxal (oxygen residues are evidenced in red), the biologically active form of xanthoxal.

**Figure 2 plants-11-01023-f002:**
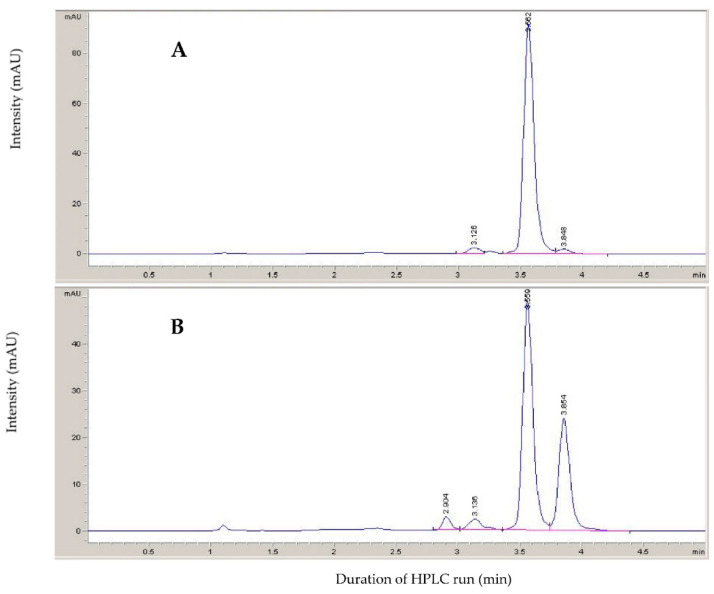
Chromatograms (at 285 nm; isocratic reversed-phase HPLC with 40% acetonitrile containing 0.02% triethylamine) of: (**A**) xanthoxal (dissolved in acetonitrile) produced by permanganate oxidation of violaxanthin; (**B**) the mixture of geometric isomers obtained after irradiation with a Wood lamp.

**Figure 3 plants-11-01023-f003:**
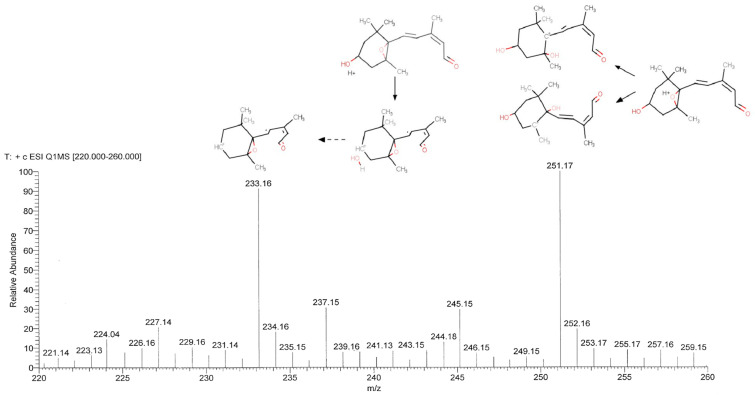
LC-ESI-MS analysis of the (unfragmented) molecule identified as xanthoxal. Two major peaks were observed, with *m*/*z* values of 251.17 and 233.16, corresponding to the protonated molecule and its dehydrated form, respectively. The putative structures (based on the *cis*,*trans*-xanthoxal isomer) above each main peak represent the proposed identification of the molecule.

**Figure 4 plants-11-01023-f004:**
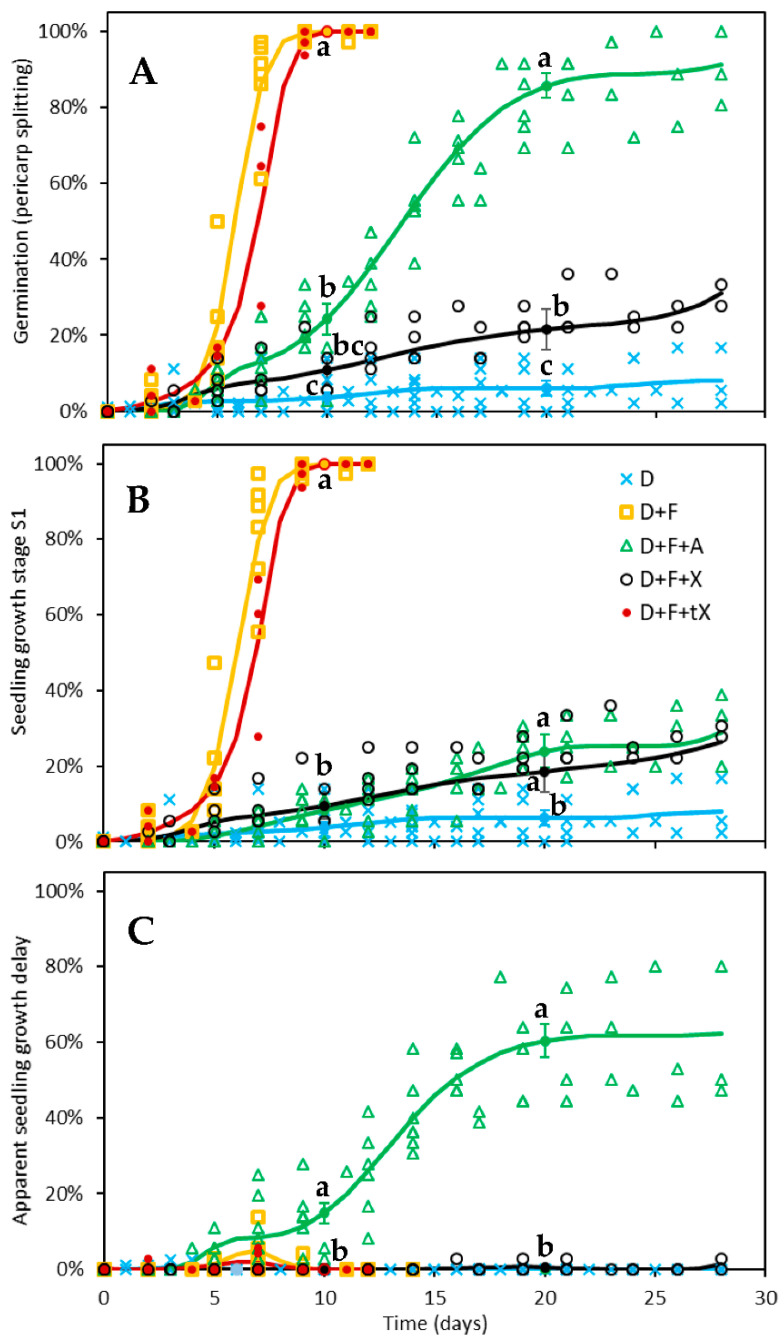
Germination and early seedling growth time-courses of dormant red rice seeds incubated in: 20 mM BTCA pH 4.4 (D, blue crosses) treated with 10 µM fluridone (D + F, yellow squares) plus either 10 µM racemic (±)-ABA (D + F + A, green triangles) or 5 µM *cis*-xanthoxal (D + F + X, black circles) or 5 µM *trans*-xanthoxal (D + F + tX, red-filled circles). Every treatment was replicated across three to six independent experimental runs, whose individual datapoints are shown in each plot. A datapoint typically represents the mean of 2–6 (typically 3) replications (12 seeds each). The number of treatment repeats (trp) and of seeds (n) tested for each treatment were as follows: D trp = 6 n = 276, D + F trp = 6 n = 204, D + F + A trp = 6 n = 251, D + F + X trp = 4 n = 144, D + F + tX trp = 3 n = 120. (**A**) Germination percentage (assessed as pericarp splitting). (**B**) Percentage of seeds attaining first growth stage (S1, i.e., rootlet or coleoptile ≥ 1 mm). (**C**) Difference between observed percentages of growth stage S1 and germination, indicative of a delay of seedling growth once germination was attained. Lines represent the time-course trends based on GzLMM estimations. Statistical differences among treatments were tested at 10 and 20 days (*p* ≤ 0.05 within each date; Holm-Sidak test): at each timepoint, treatments labelled with the same letter are not significantly different from each other (only one letter is shown for overlapping responses belonging to the same significance group). Error bars indicate standard errors.

## Data Availability

The data presented in this study are openly available in Zenodo at https://doi.org/10.5281/zenodo.6424237.

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
