# Peer review of "In Dormant Red Rice Seeds, the Inhibition of Early Seedling Growth, but Not of Germination, Requires Extracellular ABA"

_plants, 2022, doi:10.3390/plants11081023_

Round 1
Reviewer 1 Report
A reviewed manuscript is dedicated to the physiology of seed dormancy, a topic under active study. In my view, this paper is a substantial contribution to this field. A text is clearly written, although I have several questions, comments and suggestions (see attached file), mostly considering style, which may probably serve to improve this text's readability.
My may concern is also not about the scientific but style aspect. I strongly recomment to move a part of (now) 'Results and Discussion' section dedicated to the chemical synthesis and analysis of xanthoxal to the 'Materials and Methods' section. Otherwise you should list the chemical synthesis of xanthoxal among other goals of your research and probably even reflect it in a title.
Some of figure captions are also recommended to be shortened, as they contain parts of discussion.
To sum up, this paper is a fine work on physiology of seed dormancy, clearly written and finely illustrated (see several comments to the figures and their captions in manuscript file). I share a view that the main objects of review of scientific paper are style, logical argumentation, and adequacy of methods. Principal components of discussion and interpretation are a matter of readers' estimate and debates.

Reviewer 2 Report
The work tiled: In Dormant Red Rice Seeds, Inhibition of Early Seedling Growth, But Not of Germination, Requires Extracellular ABA present a broad knowledge about role of ABA in rice dormancy. During several years dormancy in rice was overcome, we get a seed without of dormancy what reflect as to fast germination, eaven before collection from the filds. Research concerns about dormancy in rice is know a very important. Is connected not only with basic knowledge about dormancy but also with practical usage. This manuscript fulfil gap in our understanding of ABA during proces of dormancy. Different hypothesis are discussed, and different point of view was shown in this papers.
Reviewer 3 Report
The article describes a well-performed study that complements existing ideas about the mechanism of regulation of seed germination and inhibition of early growth of seedlings. And the proposed technology for extracting xanthoxal after comprehensive confirmation of the quality of the resulting product can be useful for obtaining analytical standards of this substance. However, there are a number of questions and suggestions for this manuscript: - A lot of work has been done to extract and purify xanthoxal. But I believe that to confirm the structure of the xanthoxal molecule, it is not enough to use only the LC/MS method. I would like to see the results of other tests, for example 1H NMR, 13C NMR specta. - Has the purity of the xanthoxal solution been checked using wavelengths less than 285 nm for the detection of substances in chromatographic solutions? Many impurities can be detected in this way. - Why was not chemically synthesized xanthoxal or its analytical standard used as a reference substance for xanthoxal obtained from a biological source? - Specify how much xanthoxal was obtained from a unit of mass of corn leaves. THANKSAuthor Response
Please see the attachment.
